# Lymphovascular Invasion at the Time of Radical Prostatectomy Adversely Impacts Oncological Outcomes

**DOI:** 10.3390/cancers16010123

**Published:** 2023-12-26

**Authors:** Niranjan J. Sathianathen, Marc A. Furrer, Clancy J. Mulholland, Andreas Katsios, Christopher Soliman, Nathan Lawrentschuk, Justin S. Peters, Homi Zargar, Anthony J. Costello, Christopher M. Hovens, Conrad Bishop, Ranjit Rao, Raymond Tong, Daniel Steiner, Daniel Moon, Benjamin C. Thomas, Philip Dundee, Jose Antonio Rodriguez Calero, George N. Thalmann, Niall M. Corcoran

**Affiliations:** 1Department of Urology, The Royal Melbourne Hospital, The University of Melbourne, Melbourne, VIC 3050, Australiacjpmulholland@gmail.com (C.J.M.); chrissol1312@gmail.com (C.S.); lawrentschuk@gmail.com (N.L.); jspeters@netspace.net.au (J.S.P.); homi.zargar@gmail.com (H.Z.); tony@tonycostello.com.au (A.J.C.); ranjit@raourology.com.au (R.R.); raysktong@yahoo.com.au (R.T.); drdanielmoon25@gmail.com (D.M.); bcthomas_79@hotmail.com (B.C.T.); phildundee@gmail.com (P.D.); niallmcorcoran@gmail.com (N.M.C.); 2Department of Urology, University of Bern, 3010 Bern, Switzerland; andreas.katsios@insel.ch (A.K.); george.thalmann@insel.ch (G.N.T.); 3Epworth Healthcare, Melbourne, VIC 3121, Australia; cbhovens@gmail.com (C.M.H.); conradbishop@hotmail.com (C.B.);; 4Department of Urology, Solothurner Spitäler AG, Kantonsspital Olten, 4600 Olten, Switzerland; 5Department of Urology, Solothurner Spitäler AG, Bürgerspital Solothurn, 4500 Solothurn, Switzerland; 6Australian Prostate Centre, North Melbourne, VIC 3051, Australia; 7Department of Urology, Footscray Hospital Western Health, Melbourne, VIC 3011, Australia; 8Genitourinary Oncology, Peter MacCallum Cancer Centre, Melbourne, VIC 3050, Australia; 9Institute of Pathology, University of Bern, 3010 Bern, Switzerland; jose.rodriguez@unibe.ch; 10Victorian Comprehensive Cancer Centre, Melbourne, VIC 3050, Australia; 11Department of Surgery, University of Melbourne, Melbourne, VIC 3050, Australia

**Keywords:** prostate cancer, lymphovascular invasion, prostatectomy, outcomes research, prostate surgery

## Abstract

**Simple Summary:**

In this study, we investigated the impact of lymphovascular invasion at the time of prostatectomy on oncological outcomes in patients with prostate cancer. We found that the presence of lymphovascular invasion at the time of radical prostatectomy is associated with aggressive prostate cancer disease features and is an indicator of poor oncological prognosis. Therefore, lymphovascular invasion status could be used as a factor to identify patients that have underlying disseminated disease and may benefit from systemic treatment intensification, especially those with high-risk disease. Further prospective studies need to be conducted to validate these findings.

**Abstract:**

Lymphovascular invasion, whereby tumour cells or cell clusters are identified in the lumen of lymphatic or blood vessels, is thought to be an essential step in disease dissemination. It has been established as an independent negative prognostic indicator in a range of cancers. We therefore aimed to assess the impact of lymphovascular invasion at the time of prostatectomy on oncological outcomes. We performed a multicentre, retrospective cohort study of 3495 men who underwent radical prostatectomy for localised prostate cancer. Only men with negative preoperative staging were included. We assessed the relationship between lymphovascular invasion and adverse pathological features using multivariable logistic regression models. Kaplan–Meier curves and Cox proportional hazard models were created to evaluate the impact of lymphovascular invasion on oncological outcomes. Lymphovascular invasion was identified in 19% (*n* = 653) of men undergoing prostatectomy. There was an increased incidence of lymphovascular invasion-positive disease in men with high International Society of Urological Pathology (ISUP) grade and non-organ-confined disease (*p* < 0.01). The presence of lymphovascular invasion significantly increased the likelihood of pathological node-positive disease on multivariable logistic regression analysis (OR 15, 95%CI 9.7–23.6). The presence of lymphovascular invasion at radical prostatectomy significantly increased the risk of biochemical recurrence (HR 2.0, 95%CI 1.6–2.4). Furthermore, lymphovascular invasion significantly increased the risk of metastasis in the whole cohort (HR 2.2, 95%CI 1.6–3.0). The same relationship was seen across D’Amico risk groups. The presence of lymphovascular invasion at the time of radical prostatectomy is associated with aggressive prostate cancer disease features and is an indicator of poor oncological prognosis.

## 1. Introduction

Lymphovascular invasion, whereby tumour cells or cell clusters are identified in the lumen of lymphatic or blood vessels, is thought to be an essential step in disease dissemination [1]. It has been established as an independent negative prognostic indicator in a range of cancers [2]. Lymphovascular invasion is a crucial factor in the staging of testicular and penile cancer, including differentiating between T1 and T2 disease in the former [3,4]. Despite the impact of detecting lymphovascular invasion for other cancers, it has not had an important role in prostate cancer to date.

The published literature is not clear on the association between lymphovascular invasion and prostate cancer outcomes. Cheng et al. reported in their cohort study that lymphovascular invasion was associated with higher Gleason grade, pathological T stage, positive surgical margins and lymph node metastasis [5]. They also found that lymphovascular invasion was an indicator of adverse cancer outcomes. Similarly, Shariat and colleagues found that the presence of lymphovascular invasion at radical prostatectomy increased the risk of failed salvage radiation therapy, distant metastasis and death [6]. However, after adjusting for all the relevant clinical factors in multivariable analysis, lymphovascular invasion was not a significant factor in predicting biochemical recurrence. Similarly, Loeb et al. reported that although lymphovascular invasion was associated with aggressive pathological features in univariate analysis, it did not act as an independent predictors of disease progression in multivariable analysis [7]. Several other studies have also shown that lymphovascular invasion was not significantly associated with adverse oncological outcomes [8,9,10]. A meta-analysis examining the impact of lymphovascular invasion on prostate cancer outcomes reported that lymphovascular invasion increased the risk of biochemical recurrence (HR 1.5, 95%CI 1.3–1.7) but there was significant heterogeneity, with half of the included studies actually showing no impact on biochemical recurrence [11]. As such, the utility of lymphovascular invasion status is uncertain in prostate cancer.

We therefore aimed to assess the impact of lymphovascular invasion at the time of radical prostatectomy on oncological outcomes. This is the largest study in the literature to examine this question.

## 2. Materials and Methods

### 2.1. Patient Population

This was a multicentre, retrospective cohort study that included a consecutive series of patients that underwent radical prostatectomy for prostate cancer in Bern, Switzerland and Melbourne, Australia between 1994 and 2021. Surgery was performed using either an open or robotic approach. Undertaking a pelvic lymph node dissection and the extent were decided upon by the operating surgeon. We collected baseline demographic, clinical and pathological data including age, preoperative PSA value, prostatectomy Gleason score, operation approach, pathological stage, performance of pelvic lymph node dissection, surgical margin status and the presence of lymphovascular invasion.

The study was conducted in accordance with the Strengthening the Reporting of Observational Studies in Epidemiology (STROBE) statement and approved by the Ethics Committee of Canton Bern, Switzerland (KEKBE 2016-00156) and Melbourne Health (QA2020180).

### 2.2. Selection Criteria

All patients undergoing open or robotic radical prostatectomy for prostate cancer were included. We excluded patients who underwent any neoadjuvant or adjuvant therapy.

### 2.3. Staging and Follow-Up Data Collection

Preoperative staging included physical examination, measurement of the PSA, MRI of the pelvis following prostate biopsies, and either a CT scan of the thorax and abdomen and a whole-body scintigraphy to exclude bone metastases, or a PSMA PET/CT scan in clinically high-risk prostate cancer patients [12]. Follow-up investigations of patients after surgery were conducted in line with institutional protocols and international guidelines [13,14]. The data were prospectively recorded in the institutional databases. Postoperative PSA measurements and clinical examinations were taken at 3, 6 and 12 months and, afterwards, yearly. Further diagnostic imaging was obtained with a rising PSA or clinical suspicion of disease recurrence. Almost all re-staging was performed with conventional CT and whole-body bone scan because PSMA PET/CT had not yet been approved for use in the respective institutions. Patients that were staged only received either conventional staging or PSMA, not both.

### 2.4. Surgical Procedure

Every radical prostatectomy was performed or supervised by one of three senior surgeons. The decision regarding the preoperative administration of the anticoagulant agents was made on an individual patient basis [15]. Whether or not to attempt neurovascular bundle preservation was based on preoperative clinical and radiological staging and the intraoperative judgement of tumour localisation and extension [14]. The degree of attempted nerve sparing (no, uni- vs. bilateral) was judged by the surgeon and by inspection of the specimen.

### 2.5. Tissue Processing

All surgical specimens were reviewed by a specialist uropathologist in all five institutions. The excised prostate was fixed in formalin and blocked in paraffin. Transverse sections at 3.5 mm intervals were made perpendicular to the urethra from apex to base, throughout the entire specimen. Then, 5 mm sections were taken from each slice and stained with haematoxylin and eosin. The pathologist reviewing the slides determined the presence of lymphovascular invasion: tumour cells within endothelial-lined spaces with no underlying muscular walls or the presence of tumour emboli in small intraprostatic vessels. To define cancerous burden, tumour borders were outlined manually with a pen, slides were digitalised and then tumour and whole-gland volumes were calculated using image analysis software (CanoScan^®^ LiDe 70 flat-bed scanner linked to a computer running Windows^®^ XP Professional as previously described) [16].

### 2.6. Outcomes

The outcomes assessed in this study were:

Metastasis-free survival: The time from surgery to radiological visible metastasis on conventional imaging

Biochemical recurrence-free survival: The time from surgery to biochemical recurrence defined as PSA ≥ 0.2 ng/mL in two consecutive measurements [17].

### 2.7. Statistical Analysis

We reported continuous variables that are normally distributed as means and compared them using T tests. Those not normally distributed are summarised as medians and compared using a Kruskal–Wallis test. Categorical variables are reported as counts and percentages and compared using a chi-square test.

We created multivariable logistic regression models adjusting for clinical and demographic factors to assess the relationship between high-risk pathological features and lymphovascular invasion. High-risk pathological features were defined as International Society of Urological Pathology (ISUP) grade ≥4 and/or ≥pT3. Variables adjusted for included age, PSA, ISUP grade, T stage and presence of positive surgical margin.

Biochemical recurrence-free survival and metastases-free survival were analysed as a time-to-event outcome using adjusted Kaplan–Meier analyses with log-rank tests. To evaluate the effect of lymphovascular invasion at the time of RP on oncological outcomes, we created multivariable Cox proportional hazards models adjusting for relevant clinical and demographic characteristics.

We also performed subgroup analyses for all outcomes based on D’Amico risk groups. All *p* values were two-sided and 0.05 was set as the level of statistical significance. Analysis was performed in R (R Foundation for Statistical Computing, Vienna, Austria).

## 3. Results

A total of 3495 men were included in this analysis of which 19% (*n* = 653) had lymphovascular invasion present on surgical pathology. The median follow-up was 31 months [IQR 12–70]. The characteristics of the included cohort are detailed in Table 1. The median age of patients at the time of surgery was 63 years old (IQR 58–68) and they had a median PSA of 7 ng/mL (IQR 5–11).

### 3.1. High-Risk Pathology vs. Lymphovascular Invasion

There was a higher incidence of lymphovascular invasion in men with ISUP 4–5 prostate cancer compared to those with ISUP 1–3 (52% vs. 12%, *p* < 0.01, Figure 1A). Patients with ISUP ≥ 4 disease were nearly five times more likely to have lymphovascular invasion (adjusted OR 4.6, 95%CI 3.6–5.9). Similarly, there was a higher incidence of lymphovascular invasion in those with pathological T stage ≥ 3 compared to men with organ-confined disease (71% vs. 11%, *p* < 0.01, Figure 1B). Non-organ-confined disease men were more than seven times more likely to have lymphovascular invasion (OR 7.1, 95%CI 5.3–9.4).

Of the patients that had a pelvic lymph node dissection (*n* = 1328), men that had lymphovascular invasion on the surgical specimen had a higher incidence of node-positive disease compared to those that were lymphovascular invasion-negative (85% vs. 15%, *p* < 0.01). The presence of lymphovascular invasion significantly increased the likelihood of pathological node-positive disease in multivariable logistic regression analysis (OR 15, 95%CI 9.7–23.6).

### 3.2. Oncological Outcomes

The median biochemical recurrence-free survival for men that were lymphovascular invasion-positive was 20 months (IQR 15–24) and was not reached for those that were lymphovascular invasion-negative (Figure 2). The presence of lymphovascular invasion at RP significantly increased the risk of biochemical recurrence (HR 2.0, 95%CI 1.6–2.4). In the D’Amico high-risk subgroup, the median biochemical recurrence-free survival for men with lymphovascular invasion was 8 months (IQR 6–11), compared to 39 months (IQR 30–53) for those without lymphovascular invasion (Appendix A). Lymphovascular invasion significantly increased the risk of biochemical recurrence in this subgroup (HR 1.8, 95%CI 1.5–2.3). A similar impact on biochemical recurrence was seen in those with intermediate-(HR 2.8, 95%CI 2.2–3.6) and low-risk (HR 3.7, 95%CI 1.1–12.1) disease (Appendix A).

The median metastases-free survival for men that were lymphovascular invasion-positive was 127 months (IQR 93–169) and was not reached for those that were lymphovascular invasion-negative (Figure 3). Lymphovascular invasion significantly increased the risk of metastatic progression in the whole cohort (HR 2.2, 95%CI 1.6–3.0). For D’Amico high-risk patients, the median metastases-free survival for those with lymphovascular invasion was 71 months (IQR 56–92) compared to 223 months (IQR 173—not reached) for those without lymphovascular invasion (Appendix B). There was a significantly increased risk of metastases in men with D’Amico high-risk disease that were lymphovascular invasion-positive (HR 1.7, 95%CI 1.3–2.4). The same association was seen for men with D’Amico intermediate-risk disease (HR 5.2, 95%CI 3.3–8.2). Metastases-free survival analysis could not be performed for men with low-risk disease because the number of events was too low.

## 4. Discussion

The results of this study demonstrate that the presence of lymphovascular invasion at the time of radical prostatectomy is associated with aggressive prostate cancer disease features and is an indicator of poor oncological prognosis. There was a higher incidence of lymphovascular invasion in patients with a higher ISUP grade and those with non-organ-confined disease. Furthermore, 85% of patients with node-positive disease had evidence of lymphovascular invasion in their prostate specimen, suggesting that invasion of cancer into intraprostatic vessels and/or lymphatics is a precursor for disease metastasis. Most importantly, lymphovascular invasion was an indicator of poor oncological prognosis. There was a significantly increased risk of biochemical recurrence and developing metastatic disease in men that were lymphovascular invasion-positive compared to those that had no lymphovascular invasion. This has significant implications for patient counselling and, potentially, disease management.

There are several studies in the literature that support our results. A cohort of 504 men from Indiana, United States, with a median 44 months follow-up demonstrated that lymphovascular invasion significantly increased the risk of biochemical recurrence (HR 1.6, 95%CI 1.1–2.3) and also cancer-specific death (HR 2.8, 95%CI 1.0–2.3) on multivariable analysis [5]. Kang and colleagues found that 12% of patients in their radical prostatectomy cohort were lymphovascular invasion-positive and that this conferred a worse biochemical recurrence-free survival [18]. May et al. found that the 42 men with lymphovascular invasion in their prostatectomy specimen had worse 5-year biochemical recurrence-free survival than those that had no lymphovascular invasion (87% vs. 38%, *p* < 0.01) [19]. In an analysis of the National Cancer Database, Rakic et al. examined the interaction between lymphovascular invasion and lymph node invasion in those who had undergone pelvic lymph node dissection and found that patients that were both lymphovascular invasion-positive and had nodal disease had worse outcomes than those that had only lymphovascular invasion-positive or lymph node invasion [20]. They interestingly demonstrated that the survival of men that were lymphovascular invasion-positive but N0 was the same as those that were N1 but lymphovascular invasion-negative. Other studies have demonstrated that lymphovascular invasion has no impact on prognosis, but these studies generally had a lower prevalence of lymphovascular invasion-positive status, limited follow-up and had a population of lower-risk disease, meaning that the statistical power to detect a difference was limited. The majority of the literature, however, supports that lymphovascular invasion acts as an independent prognostic indicator of adverse cancer outcomes.

Our data further suggest that the presence of lymphovascular invasion can be used as an indicator of poor prognosis and as a selection criterion to identify patients that may benefit from adjuvant therapy, especially for those with high-risk disease. Lymphovascular invasion has been associated with an increased risk of nodal disease in the literature. Jiang et al. in their review reported a significant increased risk of lymph node metastases in patients that are positive for lymphovascular invasion (OR 19, 95%CI 9–44) [11]. Although we have level I evidence and the guidelines recommend the use of early salvage radiotherapy over adjuvant radiotherapy, there are emerging data that a proportion of men with high-risk disease could benefit from administering radiotherapy prior to biochemical recurrence [21,22,23]. A multi-institutional cohort of 26,118 patients reported that, at a median follow-up of 8.2 years, all-cause mortality was lower in the group of patients who had adjuvant radiotherapy compared to early salvage radiotherapy if they had adverse pathological features at radical prostatectomy, defined as high Gleason score (≥8), extraprostatic disease (≥pT3) and/or node-positive (pN1) [24]. A subsequent study demonstrated that the survival benefit was associated with increasing volume of nodal disease. A 15% increase in 7-year all-cause mortality was seen in patients with four or more lymph nodes that had undergone adjuvant radiotherapy but there was no difference in the subgroup of patients with 1–3 lymph nodes only [25]. Our study demonstrated a significantly higher incidence of pathological nodal disease in patients that are lymphovascular invasion-positive compared to those that did not have lymphovascular invasion. This suggests that lymphovascular invasion can be used as a surrogate marker for nodal disease, especially in cases where node dissection is not performed because of the variation in clinical practice globally. Therefore, the presence of lymphovascular invasion could be used to select patients who may benefit from adjuvant radiotherapy because of their increased risk of recurrence, but this needs testing in further research designed to specifically answer this question.

Furthermore, the STAMPEDE trial reported that systemic treatment with Abiraterone acetate and/or Enzalutamide with standard of care resulted in an improved overall survival compared to standard of care alone in patients with high-risk, non-metastatic prostate cancer [26]. The high-risk features that patients needed to be eligible for this trial included node-positive disease; or PSA ≥ 40 ng/mL, ISUP ≥ 4 and/or pathological stage ≥ 3 in those that were node-negative. Lymphovascular invasion was associated with each of these features in our analysis and could be used as either an additional factor for selection for adjuvant systemic treatment or as a surrogate in patients that did not undergo a pelvic lymph node dissection. Even in patients with D’Amico high-risk disease, the men that were lymphovascular invasion-positive had a significantly worse prognosis than those that were lymphovascular invasion-negative. Therefore, given the concerns about added toxicity and cost with Abiraterone acetate +/− Enzalutamide, the presence of lymphovascular invasion could act as a criterion to identify a limited subset of patients that would most benefit from additional systemic therapy. Using lymphovascular invasion status as a stratifying factor instead of pathological lymph node invasion may spare patients the morbidity of lymph node dissection for staging purposes, especially given the trend away from performing nodal dissection in current practice [27]. Jeong et al. also demonstrated that men that were lymphovascular invasion-positive were more likely to experience clinical disease progression compared to those that were lymphovascular invasion-negative [28]. This suggests that the presence of lymphovascular invasion may be indicative of underlying systemic disease and this group of patients should be considered for (systemic) treatment intensification.

The findings of this study should be interpreted within the context of its limitations. Despite being one of the largest cohorts to assess this question with long-term follow-up of important oncological outcomes, this is still a retrospective study that is subject to the shortcomings of such a design. The follow-up protocol was not uniform throughout the study and this may confound the results. However, it is unlikely that the presence of lymphovascular invasion at surgery alone impacted the decision making of clinicians post-operatively because this is not a factor that is outlined in the guidelines as being important to disease management [13]. Furthermore, there was no central pathology review, so there may be variation between the pathological analysis and interpretation of lymphovascular invasion [29]. The presence of lymphovascular invasion in our cohort was only reported as either positive or negative, but there may have been cases that were less clear; however, these have been shown to have a similar prognosis to cases that were definitely positive [30]. Furthermore, we could not define the relationship between lymphovascular invasion and higher Gleason score; nonetheless, lymphovascular invasion was shown to be a significant prognostic variable in our model. Further research should examine the importance of other pathological factors from surgical specimens that can be prognostic of long-term oncological outcomes. For example, TMPRSS2-ERG gene fusion has been associated with aggressive disease in a number of studies [31]. Commercial tests can be used to measure these molecular changes and guide prognosis [32]. Additional studies are needed to evaluate whether lymphovascular invasion remains an independent prognostic indicator when included in multivariable models with recognised molecular markers of aggressive disease.

## 5. Conclusions

Lymphovascular invasion at the time of radical prostatectomy is associated with more aggressive disease features including non-organ-confined and node-positive prostate cancer. Lymphovascular invasion is an independent prognostic indicator of inferior cancer outcomes including biochemical recurrence and development of metastatic disease. Therefore, lymphovascular invasion status could be used as a factor to identify patients that have underlying disseminated disease and may benefit from systemic treatment intensification, especially those with high-risk disease. Further prospective studies need to be conducted to validate these findings.

## Figures and Tables

**Figure 1 cancers-16-00123-f001:**
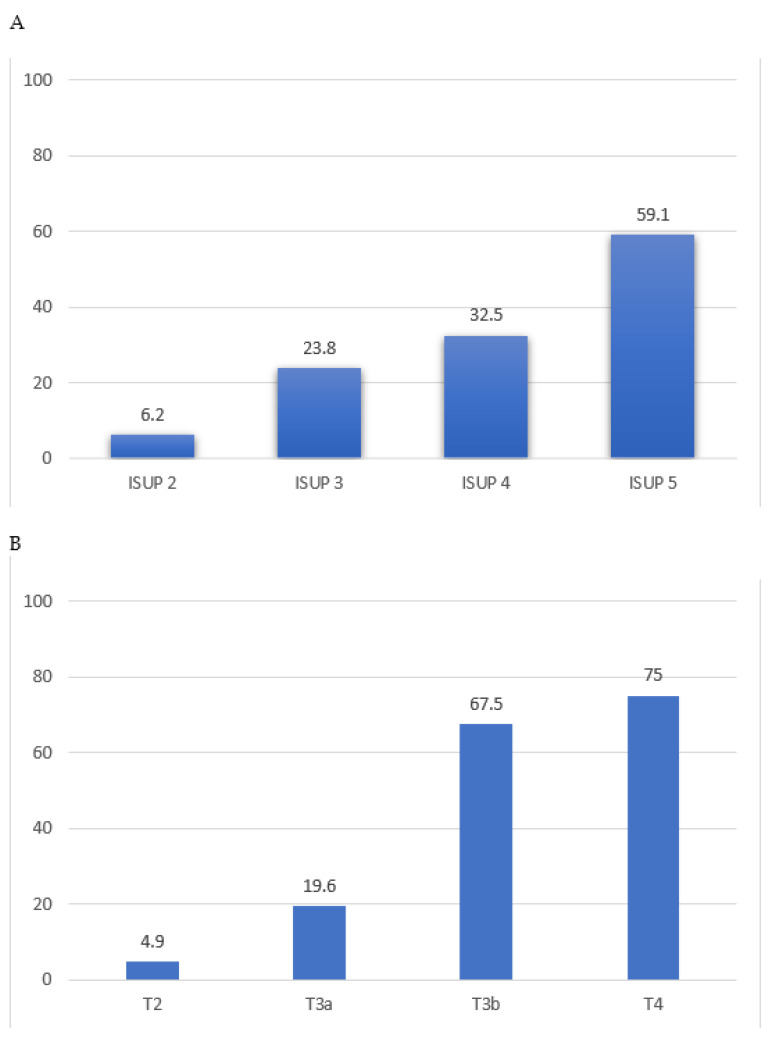
(**A**) ISUP grade vs. lymphovascular invasion and (**B**) pathological T stage vs. lymphovascular invasion.

**Figure 2 cancers-16-00123-f002:**
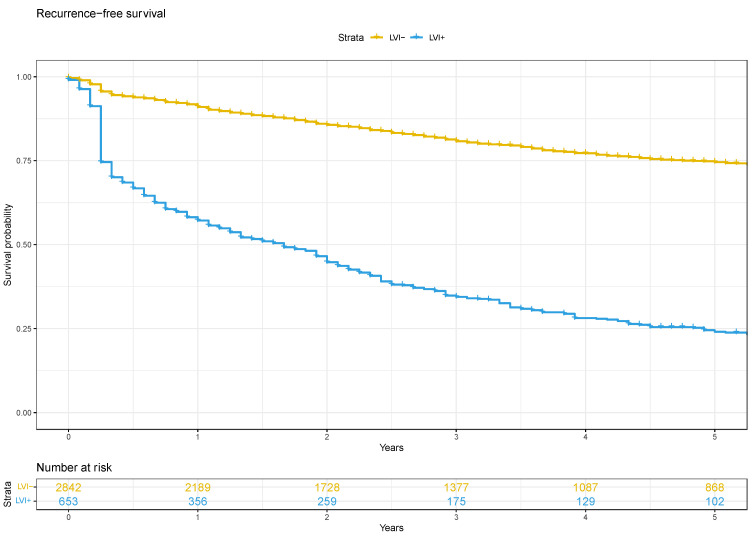
Adjusted Kaplan–Meier curve for biochemical recurrence-free survival.

**Figure 3 cancers-16-00123-f003:**
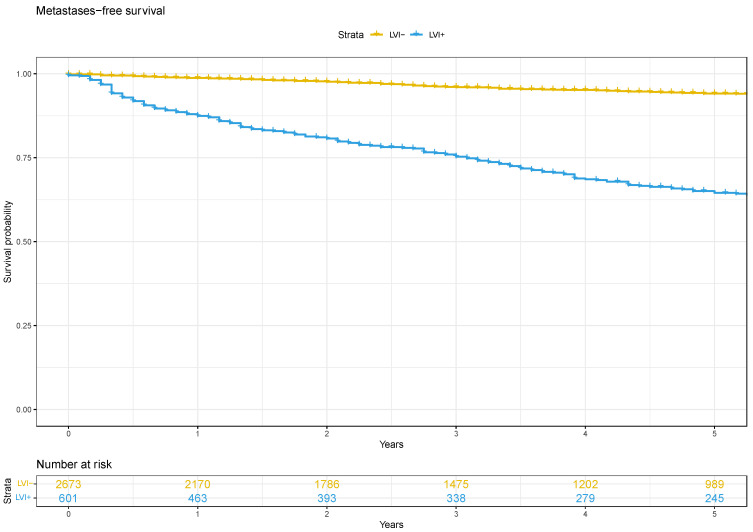
Adjusted Kaplan–Meier curve for metastasis-free survival.

**Table 1 cancers-16-00123-t001:** Cohort characteristics.

	Lymphovascular Invasion-Negative (*n* = 2842)	Lymphovascular Invasion-Positive (*n* = 653)	*p*-Value
Age (years), median (IQR)	63 (58–68)	65 (60–69)	<0.01
PSA (ng/mL), median (IQR)	7 (5–10)	12 (7–20)	<0.01
ISUP, *n* (%)			<0.01
1	545 (19%)	63 (9.7%)	
2	1415 (50%)	127 (19%)	
3	603 (21%)	162 (25%)	
4	152 (5.4%)	81 (12%)	
5	119 (4.2%)	219 (34%)	
Pathological T stage 3–4, *n* (%)	128 (4.5%)	312 (48%)	<0.01
Tumour volume (cc), median (IQR)	3 (1–8)	12 (6–28)	<0.01
PSM, *n* (%)	744 (26%)	338 (52%)	<0.01
PLND, *n* (%)	1037 (64%)	422 (84%)	<0.01
N+, *n* (%)	45 (4.9%)	247 (62%)	<0.01

## Data Availability

Data are available on request due to restrictions (privacy and ethical).

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
