# Peer review of "Lymphovascular Invasion at the Time of Radical Prostatectomy Adversely Impacts Oncological Outcomes"

_cancers, 2023, doi:10.3390/cancers16010123_

Round 1
Reviewer 1 Report
Comments and Suggestions for Authors
The authors present a study in which they investigated the impact of lymphovascular invasion at the time 24 of prostatectomy on oncological outcomes in patients with prostate cancer. In congratulate the authors to this fantastic work which adds scientific value. Here are some of my comments / questions:
1. Only patients with negative pre-operative staging were included. In my understanding, only patient at higher risk get pre-op staging (e.g. higher Gleason score, high PSA, high number of positive biopsies). Don't you think this could be a bias for this study as a preselection of patients?
2. As you state in association with Cheng et al. (page 2, line 63), do you think LVI positivity is more or less a surrogate for known risk factors such as higher Gleason, T-stage, R positivity? Do we really need to consider LVI? To answer this question you may have a look at Table 1 that shows this huge discrepany in ISUP grading between both cohorts.
3. In M&M section you describe the different staging modalities used. However, it is not clear to me whether all patients received either CT and szintigraphy or PSMA or whether some patient had only CT no szintigraphy or vice versa.
4. I think a link to molecular differences in prostate cancer patients of higher risk would be great. Looking at data from ASCO 2023 (Reig, Aparichio #5083) it gets clearer that molecular changes might be driver for more aggressive disease (e.g. two of three mutations at PTEN, RP1, TP53). Also patients with BCRA 1/2 mutations may have worse outcome. Maybe you add a little statement in your discussion.
Author Response
Reviewer #1
The authors present a study in which they investigated the impact of lymphovascular invasion at the time of prostatectomy on oncological outcomes in patients with prostate cancer. In congratulate the authors to this fantastic work which adds scientific value. Here are some of my comments / questions:
1.
Only patients with negative pre-operative staging were included. In my understanding, only patient at higher risk get pre-op staging (e.g. higher Gleason score, high PSA, high number of positive biopsies). Don't you think this could be a bias for this study as a preselection of patients?
We thank the author for this comment and agree that we only included patients that were eligible for radical prostatectomy but contend that this is not introducing a significant bias because the aim was to assess the impact of LVI on outcomes after radical prostatectomy specifically rather than all prostate cancer patients.
2.
As you state in association with Cheng et al. (page 2, line 63), do you think LVI positivity is more or less a surrogate for known risk factors such as higher Gleason, T-stage, R positivity? Do we really need to consider LVI? To answer this question you may have a look at Table 1 that shows this huge discrepany in ISUP grading between both cohorts.
We contend that LVI is an important factor as our multivariable models adjusted for PSA, Gleason score, T stage and positive margins and yet LVI was shown to be an important factor.
3.
In M&M section you describe the different staging modalities used. However, it is not clear to me whether all patients received either CT and szintigraphy or PSMA or whether some patient had only CT no szintigraphy or vice versa.
We have now clarified this in the Methods: “Patients that were staged only received either conventional staging or PSMA, not both.”
4.
I think a link to molecular differences in prostate cancer patients of higher risk would be great. Looking at data from ASCO 2023 (Reig, Aparichio #5083) it gets clearer that molecular changes might be driver for more aggressive disease (e.g. two of three mutations at PTEN, RP1, TP53). Also patients with BCRA 1/2 mutations may have worse outcome. Maybe you add a little statement in your discussion.
We thank the author for this suggestion and made the following addition in the Discussion: “Further research should examine the importance of other pathological factors from sur-gical specimens that can be prognostic of long-term oncological outcomes. For example, TMPRSS2-ERG gene fusion has been associated with aggressive disease in number of studies(26). Commercial tests can be used to measure these molecular changes and guide prognosis(27).”
Reviewer 2 Report
Comments and Suggestions for Authors
This is a large rertospective study about the potential impact of LVI in oncological outcomes after RP. The study is well presented but I have some remarks
1. There are enough evidence that clearly support that the presence of LVI is a bad prognostic factor. In my point of view another study (and a retrospective one) does not add something to the existing level of evidence
2.Authors state that the median follow up of the patients is 3 years. First of all in a cohort with patients from 1994 I dont understand why authors reduce the follow up period so significantly and in the same time they diminish the strength of their study. Secondly this 3 year period is very small for onbcological outcomes of prostate cancer. I also dont understand how on a follow up period of 3 years authors present statistics with 5+years (even if the range is up to 7 years the statistics would have been severely biased).
3.In order to check the role of one factor (LVI) to the final oncological outcomes it would be wise to have matched paired analysis and the two groups must be matched in terms of other factors that may affect final outcomes. We see that the groups are with statistically significant completely different. In my point of view a multivariate analysis cannot address this importat bias.
Author Response
Reviewer #2
This is a large rertospective study about the potential impact of LVI in oncological outcomes after RP. The study is well presented but I have some remarks
1.
There are enough evidence that clearly support that the presence of LVI is a bad prognostic factor. In my point of view another study (and a retrospective one) does not add something to the existing level of evidence
We thank the reviewer for their comment but contend that this study provides a large dataset with granular details and long term oncological outcomes that is not seen in other studies on this subject in the literature.
2. Authors state that the median follow up of the patients is 3 years. First of all in a cohort with patients from 1994 I dont understand why authors reduce the follow up period so significantly and in the same time they diminish the strength of their study. Secondly this 3 year period is very small for oncological outcomes of prostate cancer. I also dont understand how on a follow up period of 3 years authors present statistics with 5+years (even if the range is up to 7 years the statistics would have been severely biased).
The 3 year follow-up represents the median follow-up so there is 50% of the cohort who had longer follow-up and we have presented this data because 40% of the patients still have data at 5 years so contend that this is not overly biased.
3. In order to check the role of one factor (LVI) to the final oncological outcomes it would be wise to have matched paired analysis and the two groups must be matched in terms of other factors that may affect final outcomes. We see that the groups are with statistically significant completely different. In my point of view a multivariate analysis cannot address this important bias.
We thank the reviewer for their comment and did consider this but did not want to exclude large proportion of the cohort and reduce the power. We believe that multivariable model adjusts for these differences and there is no significant difference by employing propensity score matching as discussed in the literature: https://pubmed.ncbi.nlm.nih.gov/21616172/
https://pubmed.ncbi.nlm.nih.gov/17960554/
This is also supported in the Urology Statistical guidelines (https://www.ncbi.nlm.nih.gov/pmc/articles/PMC6397060/): “A common assumption is that propensity methods somehow provide better adjustment for confounding than traditional multivariable methods. Except in certain rare circumstances, such as when the number of covariates is large relative to the number of events, propensity methods give extremely similar results to multivariable regression.”
Reviewer 3 Report
Comments and Suggestions for Authors
This article considers the relationship between lymphovascular invasion and prognosis in prostate cancer.
Patients with lymphovascular invasion have higher Greason Grade and higher risk, but it is difficult to determine whether it is a cause or a result of the invasion.
It may be possible to evaluate the prognosis of patients with lymphovascular invasion by considering the prognosis of low grade or low stage patients with lymphovascular invasion.
Since there are evaluating a long period of time, the treatment outcome may be influenced by the time period.
There is an extra space between words.
2.6. Outcome
Biochemical recurrence-free survival: the time from surgery to biochemical recurrence__was defined as ...
Author Response
Reviewer #3
This article considers the relationship between lymphovascular invasion and prognosis in prostate cancer.
Patients with lymphovascular invasion have higher Greason Grade and higher risk, but it is difficult to determine whether it is a cause or a result of the invasion.
We agree that there is a higher risk of LVI with increasing Gleason grade and from our study we cannot determine whether it is a cause or result but regardless on multivariable adjustment, LVI was shown to be an important factor and have acknowledged this in the Discussion: “Furthermore, we could not define the relationship between LVI and higher Gleason score but nonetheless, LVI was shown to be a significant prognostic variable.”
It may be possible to evaluate the prognosis of patients with lymphovascular invasion by considering the prognosis of low grade or low stage patients with lymphovascular invasion.
We were unable to do this and reported as such in the Results because of the low number of events as expected for this disease risk.
Since there are evaluating a long period of time, the treatment outcome may be influenced by the time period.
We thank the reviewer for this comment and have adjusted for this.
There is an extra space between words.
This objection is greatly appreciated. We have corrected it accordingly.
2.6. Outcome
Biochemical recurrence-free survival: the time from surgery to biochemical recurrence__was defined as ...
Thanks for noticing this error – this has been corrected.
Reviewer 4 Report
Comments and Suggestions for Authors
The manuscript by Sathianathen demonstrated that LVI at the time of radical prostatetomy was associated with higher pathological grade as well as worse oncological outcome, even in the high risk group.
The major question that needs to be clarified is that if LVI an independent predictor of worsening oncological outcome. The author described in line 145 "multivariable logistical regression models adjusting for clinical and demographic factors". The variables included in the model need to be clearly described with their p values listed.
Authors also mentioned in the introduction that some prior published studies (Reference 7-10) did not establish LVI as an independent factor to predict adverse outcome. Why did that happen? What was the difference between those studies and the current study? Some discussion should be included in the manuscript.
Author Response
Reviewer #4
The manuscript by Sathianathen demonstrated that LVI at the time of radical prostatetomy was associated with higher pathological grade as well as worse oncological outcome, even in the high risk group.
The major question that needs to be clarified is that if LVI an independent predictor of worsening oncological outcome. The author described in line 145 "multivariable logistical regression models adjusting for clinical and demographic factors". The variables included in the model need to be clearly described with their p values listed.
We have included the variables adjusted for in the multivariable model: “Variables adjusted for included age, PSA, ISUP grade, T stage and presence of positive surgical margin.”
However, we have not included all the variables and P values as per the Statistical Guidelines: https://pubmed.ncbi.nlm.nih.gov/30537407/
“In a typical observational study, an investigator might explore the effects of two different approaches to radical prostatectomy on recurrence while adjusting for covariates such as stage, grade and PSA. It is rarely worth reporting estimates such as odds or hazard ratios for the covariates. For instance, it is well known that a high Gleason score is strongly associated with recurrence: reporting a hazard ratio of say, 4.23, is not helpful and a distraction from the key finding, the hazard ratio between the two types of surgery.”
Authors also mentioned in the introduction that some prior published studies (Reference 7-10) did not establish LVI as an independent factor to predict adverse outcome. Why did that happen? What was the difference between those studies and the current study? Some discussion should be included in the manuscript.
We thank the reviewer for their comment and have now discussed this in the manuscript: “Other studies have demonstrated that LVI has no impact on prognosis but these studies generally had a lower prevalence of LVI positive status, limited follow-up and had a population of lower-risk disease meaning that the statistical power to detect a difference was limited.”
This retrospective study demonstrates the correlation of tumor aggressiveness with lymphovascular invasion.
Particularly in terms of poor prognosis. This is a central issue for the management of follow-up after radical prostatectomy.
Numerous other works over the years have already widely corroborated what is reported in this study. Already in 2005 Cheng et al. (references 5) with a population of over 500 patients had demonstrated lymphovascular involvement in 21% of the patients examined, demonstrating how lymphovascular involvement was an independent risk factor for PSAtot.
All re-staging was performed with conventional CT and whole-body bone scan and not with the more recent PSMA PET/CT. This represents a disgraceful limit for correct staging in the event of biochemical disease recurrence as reported in the most important international guidelines.
We thank the reviewer for this comment but unfortunately PSMA is even still not widely available in the world and conventional imaging is still used for staging. We do not believe that this would have a significant impact.
Reviewer 5 Report
Comments and Suggestions for Authors
This retrospective study demonstrates the correlation of tumor aggressiveness with lymphovascular invasion.
Particularly in terms of poor prognosis. This is a central issue for the management of follow-up after radical prostatectomy.
Numerous other works over the years have already widely corroborated what is reported in this study. Already in 2005 Cheng et al. (references 5) with a population of over 500 patients had demonstrated lymphovascular involvement in 21% of the patients examined, demonstrating how lymphovascular involvement was an independent risk factor for PSAtot.
All re-staging was performed with conventional CT and whole-body bone scan and not with the more recent PSMA PET/CT. This represents a disgraceful limit for correct staging in the event of biochemical disease recurrence as reported in the most important international guidelines.
Author Response
Thank you for taking the time to review the manuscript.
Round 2
Reviewer 2 Report
Comments and Suggestions for Authors
Unfortunately the answers of the authors did not improve the manuscript
Reviewer 4 Report
Comments and Suggestions for Authors
All questions have been addressed by the authors. No additional comments.